# Investigation of Bamboo Fibrous Tensile Strength Using Modified Weibull Distribution

**DOI:** 10.3390/ma15145016

**Published:** 2022-07-19

**Authors:** Yalew Dessalegn, Balkeshwar Singh, Aart W. van Vuure, Irfan Anjum Badruddin, Habtamu Beri, Mohamed Hussien, Gulam Mohammed Sayeed Ahmed, Nazia Hossain

**Affiliations:** 1Department of Mechanical Engineering, Program of Mechanical Design and Manufacturing Engineering, Adama Science and Technology University, Adama 1888, Ethiopia; yalewdesu@yahoo.com (Y.D.); habtamu_beri@yahoo.com (H.B.); or drgmsa786@gmail.com (G.M.S.A.); 2Campus Group T, Composite Materials Group, Department Materials Engineering, Katholieke Universiteit Leuven (KU Leuven), Andreas Vesaliusstraat 13, 3000 Leuven, Belgium; aartwillem.vanuure@kuleuven.be; 3Mechanical Engineering Department, College of Engineering, King Khalid University, Abha 61421, Saudi Arabia; magami.irfan@gmail.com; 4Department of Chemistry, Faculty of Science, King Khalid University, Abha 61413, Saudi Arabia; mhalmosylhy@kku.edu.sa; 5Pesticide Formulation Department, Central Agricultural Pesticide Laboratory, Agricultural Research Center, Dokki, Giza 12618, Egypt; 6Center of Excellence (COE) for Advanced Manufacturing Engineering, Program of Mechanical Design and Manufacturing Engineering, Adama Science and Technology University, Adama 1888, Ethiopia; 7School of Engineering, RMIT University, Melbourne, VIC 3001, Australia; bristy808.nh@gmail.com

**Keywords:** bamboo fiber, fiber strength, gauge length, Weibull distributions, Weibull modulus

## Abstract

Ethiopia has a large coverage of bamboo plants that are used for furniture making and house building. So far, researchers have not studied the strength of Ethiopian bamboo fibers, which are utilized for composite applications. The current study measured the strength of bamboo fibers based on various testing lengths and calculated the predictive tensile strength using a modified Weibull distribution. Moreover, the quality of the extraction machine is evaluated based on shape and sensitivity parameters. This research paper incorporates the coefficient of variation of the fiber diameters, considering the defects distribution through the length for measuring the predictive strength of the fibers. The fiber diameters were calculated using the area weight methods, which had its density measured using a Pycnometer. It has been examined that as the testing gauge length and coefficient variation of fiber diameter simultaneously increased, the tensile strength of the bamboo fibers decreased. The shape parameter, sensitivity parameter, and characteristic strength of Injibara bamboo (*Y. alpina*) are 6.02–7.83, 0.63, and 459–642 MPa, whereas Kombolcha bamboo (*B. oldhamii*) are 5.87–10.21, 0.33, and 408–638 MPa, as well as Mekaneselam bamboo (*Y. alpina*) are 5.86–9.63, 0.33 and 488–597 MPa, respectively.

## 1. Introduction

Bamboo plants are categorized in the *Graminea* family’s *bambusae* genus, overgrowing and reaching annually [1]. Researchers have investigated the properties of bamboo fibers, including their mechanical behaviors, chemical composition, and thermal properties, making them adaptable for the composite industry. However, the type of species, growth location, harvesting season, and type of extraction affect the properties of natural fibers [2]. Previous research has investigated the mechanical behaviors of bamboo fiber and its composites in depth. There have been significant improvements in bamboo fiber-reinforced composites during the last century [3]. The bamboo plant has a multi-hierarchical complex structure composed of bamboo fiber, elementary fiber, and nanofibrils [4]. The research of its microstructure, in particular, is justified as a viable method for explaining the strength of bamboo fiber [5]. Unfortunately, most structural and strength studies of bamboo fibers have been done at the macroscopic level [6]. Researchers showed that the strength model of the fibers described the strength of bamboo fibers based on its microstructure [7]. Indeed, large amounts of fiber breakage dominate the failure to function catastrophically for fiber-reinforced composites [8]. Fibers typically exhibit strength variability because defects are distributed through the length. The statistical strength distribution of bamboo fiber is critical for designing composite structures safely. The tensile strength of the jute fiber decreased as the fiber diameter variation increased. The modified Weibull distribution predicted strength more accurately than the two conventional models. Moreover, jute fiber breaking strength was less sensitive for gauging length than cotton fiber breaking strength, since jute filament breaking involves ultimate cells breaking repeatedly and the matrix cracking [9]. Chemically retted single bamboo fiber had a 47.6% higher tensile strength than mechanically retted fibers. Tensile strength, tensile modulus, and elongation of fiber bundles were 68.8, 52.3, and 60.9% lower than that of single fibers for mechanical retting, respectively. The tensile strength of chemically retted fiber bundles was more than 2.1 times greater than that of mechanically retted fiber bundles, while the tensile modulus was nearly 1.4 times greater [10].

Bamboo fibers aged 0.5 to 8.5 years meet the requirements for use in fiber-reinforced composites. They had an average tensile strength of 1.54 GPa and a longitudinal MOE of 33.86 GPa, respectively [11]. Moso bamboo fibers have an average longitudinal tensile modulus and tensile strength of 32 to 34.6 GPa and 1.43 to 1.69 GPa, respectively [12]. The average tensile strength, elastic modulus, and strain to failure were 523.20 ± 111.65 MPa, 22.27 ± 6.29 GPa, and 3.08 ± 1.57%. From the inner part to the outer part of the bamboo stem, the tensile strength of bamboo fibers slightly tended to increase on the whole, while the tensile elastic modulus and the strain to failure had an increasing tendency and decreasing tendency, respectively [13]. Bamboo fiber is considered a significant alternative reinforcement to synthetic fibers, with significant potential for use in functional fibers composites [14]. The strength of natural fibers has higher inconsistency within fibers and between fibers, however, it is a primary consideration for industrial applications. Extraction methods, types of species, and processing parameters influenced the mechanical strength of the fibers in which defects are developed [15]. Natural fibers have a greater variety within a fiber and between a fiber diameter compared to synthetic fibers [16]. Since the strength of the fiber varies with the diameter of the fiber, it is difficult to assign the exact value of fiber strength based on fiber diameter. This necessitates the utilization of an effective way of evaluating fiber strength and predicting its size dependence. The modified Weibull distribution, a practical model with only three parameters, adequately defined the fiber strength distribution at various testing lengths. The average fiber strength is reduced as the testing length is increased, from 943 MPa at L = 1 mm to 733 MPa at L = 40 mm. However, it was not dependent on the mean fibers volume. The Weibull shape parameter for all tested fibers was found to be 7.6, indicating low strength variability in comparison to other natural fibers and some synthetic fibers, clearly showing high quality [16]. Weibull was the first to recommend a mathematical method to analyze the mechanical behavior of brittle materials that was widely applicable. The weak-link theory, which is based on Weibull statistics, is commonly used to examine the fracture strength of many brittle materials, including carbon and glass fibers [17]. In recent decades, it has also been used on natural fibers, such as hemp, flax, and jute [9]. The weakest link statistics are usually applied to express the strength of brittle fiber, which assumes that a substance is composed of smaller components linked together. If any of these elements, or “links,” fail, the substance is recognized to have failed [18]. This statistical variability is frequently examined based on the Weibull cumulative formulation equation application, which is written as follows:(1)$$P=1-\exp\left\{-{\left(\frac{V}{\mathrm{Vo}}\right)}^{\gamma }\;{\left(\frac{\sigma }{\sigma o}\right)}^{\beta }\right\}\;,$$
where P is the failure probability, β represents the shape parameter, γ is the volume sensitivity, σ_O_ represents characteristic strength that corresponds to reference volume Vo, and V represents the fiber volume at stress less than or equal to σ. For a constant measured volume, Equation (1) is simplified to
(2)$$P=1-\exp\left\{-{\left(\frac{\sigma }{\sigma o}\right)}^{\beta }\right\}\;.$$

The observations can be used to estimate the unknown cumulative probability of failure P. n is the number of data points and i is the rank of the ith data point. Then, the expectation of σ results of the proposed approximation technique is an empirical prediction of P [4]:(3)$$P=\left(\frac{i}{n+1}\right).$$

In Equation (2), we take the double natural logarithms of both sides. The following equations result from rearranging the Weibull distribution [19].
(4)$$\ln\;\left(\ln\left(\frac{1}{1-p}\right)\right)=\beta \ln\left(\sigma \right)-\;\beta \ln\left(\sigma o\right).$$

As a result, the characteristics of strength and Weibull modulus can be derived from a plot of Y = $\ln\;\left(\ln\left(\frac{1}{1-p}\right)\right)$ versus X = ln(σ). Although the conventional Weibull model is used to describe uniform brittle and polymeric fibers, it does not demonstrate fiber strength depending on fiber volume. Therefore, a modified Equation (1) was recommended by Waston and Smith [20,21].
(5)$$p=1-\;\exp[-{\left(\frac{L}{\mathrm{Lo}}\right)}^{\gamma }{\left(\frac{\sigma }{\sigma o}\right)}^{\beta }].$$

The weak-link statistics are used to derive a material’s strength, as longer fibers have more links than shorter fibers and are more likely to encounter a more severe flaw in the length of the fibers [9]. The fracture strength of shorter fibers is lower compared to longer fibers [22]. As a consequence, predictions for weak-link scaling indicate that any fiber’s strength can be scaled [23]. Based on Equation (5), it can be changed as follows:(6)ln[−ln(1−p)]=γlnL−γlnLo+βlnσ−βlnσo.

Hence, the shape parameter and characteristic strength can be observed from a plot of ln[−ln(1−p)] versus ln(σ), which is a gradient (β) and intercept σo at ln[−ln(1−p)] = 0 [24].

According to Waston, the sensitivity parameter (γ) describes variations in fiber diameter [20]. Zhang and Wang investigated the tensile strength of wool fiber using the modified Weibull distribution [25].

The purpose of this research is to investigate the tested length effects on the measured and predictive strength by calculating the shape parameter (β), sensitivity parameter (γ), and characteristic strength of Ethiopian bamboo species based on the developed extraction machine. The extraction of bamboo fibers is difficult and challenging work. The extraction machine of bamboo fibers is novel and developed by the authors in the workshop. The paper frames are prepared based on the tested length for straightening and by easily gripping the fibers to the jaw. The fiber strengths at various tested lengths were measured using the ultimate tensile strength machine (UTS) according to the ASTM standards. The quality of the fibres extraction method is assessed based on the shape parameter, sensitivity parameter, and characteristic strength. Ethiopia has the largest coverage of bamboo plants in Africa, and bamboo culm in Ethiopia is used in furniture making, house building, and fencing. It is applied for low-level application of utilization. However, so far, researchers have not studied the ultimate tensile strength of bamboo fibers in Ethiopia. Therefore, the authors want to determine the quality of the extraction of bamboo fibers based on fiber strength distribution using modified Weibull strength distribution.

## 2. Experimental Details

### 2.1. Description of the Study Area

The study region, location, and weather conditions of the testing sites are presented in Table 1. From the highest to the lowest altitude in the sea level are Mekaneselam, Injibara, and Kombolcha, whereas in annual temperature is Kombolcha, Injibara, and Mekaneselam, respectively.

### 2.2. Harvesting of the Bamboo Culm

Injibara (*Y. alpina*), Kombolcha (*Bambusa oldhamii*), and Mekaneselam (*Y. alpina*) were harvested in November 2020. Three bamboo culm samples were harvested at the age of two years in each region. Experienced field personnel determined the age of the bamboo culm based on its color, as well as sheath. The culm height was divided into bottom, middle, and top parts, which are based on an internode length greater than 25 cm and an outer diameter greater than 3 cm. Figure 1a–c depict images of Ethiopian bamboo species. Injibara and Mekaneselam bamboo plants are known as *Y. alpina*, which has the highest culm internodal length. Ethiopia has various types of bamboo species, The Kombolcha bamboo plant is known as *Bambusa oldhamii*, which is available in temperate regions. It has short internodal length and larger culm diameter.

### 2.3. Microscopy Structures of Ethiopian Bamboo Species

Each bamboo species was cut, resulting in a thickness with a dimension of 1.5 cm × 1.5 cm width and embedded in a Buehler Epoxi Cure 2 Epoxy mounting resin and Epoxi Cure 2 hardener. Sandpaper of grit sizes P80 to P400 was used to grind the specimen, which was then polished using a diamond paste with particles of 6 μm, 3 μm, and 1 μm, respectively. Diamond paste is used to polish various polishing cloths. The Lam Plan 2TS3 cloth was used in the first step, the DP-Nap cloth in the second and third steps, and the OP-Chem cloth in the final step. A light microscope (Axioskop 40, Zeiss) was used to examine the structure and type of fiber bundles existing in various bamboo species in Ethiopia.

### 2.4. Extraction of Continuous Bamboo Fiber

As shown in Figure 2, the extraction process of bamboo fibers is discussed in detail. Bamboo culms were cut at the node, then the internodes were split longitudinally into 30 mm. The strips were then passed under light pressure and at a slow speed through the rolling mill between two steel rods. After rolling sufficiently, one-third of the inner and outer parts were removed, and the middle portions were combed with various sizes of comb to extract and clean the fibers. The extracted fibers were sun-dried for two weeks, and their lengths exceeded 250 mm [26]. The fibers were extracted from culm internode numbers 9 and 13. The cleaning process was carried out, which entails brushing the fibers with water to remove residual parenchyma [27,28]. Finally, the fibers were dried at 60 °C for 72 h before producing composites.

### 2.5. Tensile Test

Before tensile measuring, we inspected the fibers visually to ensure that there were no defects along their length. Two ends of the testing samples were joined with cyanoacrylate adhesive onto a paper frame and fixed on the two ends of the frame on the machine jaw, then we cut the paper frame using scissors before running the machine. Figure 3 depicts the paper frame preparation procedure based on the tested length. The paper frame is used to keep the straightness of the fiber within the frame. A total of 15 mm, 25 mm, 30 mm, 40 mm, and 50 mm of tested length were used to measure the tensile strength and predictive strength of Ethiopian bamboo species. For each test length, 40 samples of fiber were tested, and 200 samples were tested for 1 bamboo species. Moreover, 600 samples of tests were performed for all bamboo species. Tensile measurements were carried out on the ultimate tensile strength machine of 5943 equipped with a 100 N load cell in a conditioned environment at 50% RH and 21 °C under the ASTM C1557-14 [29]. The frame was gripped pneumatically with a gripping force of 200 N and a load cell of 100 N. To straighten the fiber, a crosshead displacement rate of 1.5 mm/min, and a pre-load of 0.01 N were applied. Samples were underloaded until failure happened. As presented in Figure 4, technical bamboo fibers connect many elementary fibers that determine the mechanical strength. As the tested length of the fibers increases, the likelihood of a weak link formation increases, with the fibers breaking at a specific joint [19,30]. As a result, the strength increases, and a different fracture type can occur, particularly at different tested lengths.

### 2.6. Bamboo Fibers Cross-Section Area Measuring

The fiber cross-section of the tensile-tested fibers was evaluated using a gravimetric method. The area can be determined using the weight and density of fibers. A digital balance (Mettler AT 261 Delta Range, Mettler Toledo) with an accuracy of 0.00001 g was used to measure the weight of the fibers. The diameter of the fiber was calculated from the area by assuming a circular shape.
(7)$$A=\frac{m}{\rho l}\;,$$
where *m* represents fiber mass, *ρ* represents fiber density, and *l* represents fiber length [31].

## 3. Results and Discussions

### 3.1. Microstructure of Specimens

The bamboo fiber cross-section is observed using the light microscope presented in Figure 5a–c. Four basic types of vascular bundles were found by Liese et al. [32], and the considered species can be assigned as follows: Injibara and Kombolcha bamboo vascular bundles were of types (I), whereas Mekaneselam bamboo vascular bundles were of types (III), as indicated in Figure 5a–c. Figure 5 shows a typical polished cross-section of bamboo fiber in Ethiopia. It appears irregular, but is approximately circular. One bamboo fiber contains many elementary fibers. The cross-section of these fibers is either pentagonal or hexagonal, and it is arranged in a honeycomb pattern [33].

### 3.2. Description of the Fiber Diameters

The fiber diameter distribution of all the tested fibers is presented in Figure 6 (the diameter is calculated from the area by assuming a circular shape), also distinguishing between a thick and a thin fraction. The fiber diameters of Injibara, Kombolcha, and Mekaneselam were 100–500 μm, 50–500 μm, and 100–450 μm, respectively. The highest population of fiber diameters in Injibara, Kombolcha, and Mekaneselam bamboo were 250–300 μm, 200–250 μm, and 350–400 μm, respectively. Kombolcha bamboo fibers had the lowest measured fiber diameter from the fiber diameter population, as shown in Figure 6.

### 3.3. Determination of Fiber Density

The fiber densities of Injibara, Kombolcha, and Mekaneselam were 1.35–1.38 g/cm^3^, 1.35–1.36 g/cm^3^, and 1.34–1.36 g/cm^3^, respectively. Injibara (*Y. alpina*) has a 1.4% higher fiber density compared to Kombolcha (*B. oldhamii*) and Mekaneselam (*Y. alpina*), whereas similar fiber densities were registered in Kombolcha and Mekaneselam. Ethiopian bamboo fibers (Injibara, Kombolcha, and Mekaneselam) have a lower fiber density compared to *Guadua angustifolia Kunth* (GAK) (1.41 ± 0.03 g/cm^3^), *Dendrocalamus membranaceus Munro* (DMM) (1.41 ± 0.02 g/cm^3^), *phyllostachys aureosulcata Spectabilis* (PAS) (1.43 ± 0.02 g/cm^3^), PV (1.44 ± 0.03 g/cm^3^), and *Phyllostachys vivax McClure Jiantonging* (PVMJ) (1.47 ± 0.01 g/cm^3^). However, they have a similar fiber density compared to *phyllostachys nigra Boryana* (PNB) (1.38 ± 0.01 g/cm^3^) [34].

### 3.4. Correlation with Within-Fiber Diameter Variations

According to Zhang, the sensitivity parameter (γ) is calculated by plotting the slope of the logarithm of CVFD versus the logarithm of the tested length [28]. Figure 7 describes the relation between CVFD and the tested Mean length. The coefficient value R-square of Injibara, Kombolcha, and Mekaneselam bamboo fibers was 97%, 96%, and 93%, respectively. This percentage indicates that the linear fit between the logarithm value of CVFD and the tested length is satisfactory. The sensitivity parameter (γ) of Injibara, Kombolcha, and Mekaneselam was 0.63, 0.33, and 0.33, respectively. The equation formula of the sensitivity parameter (γ) of Injibara fibers was ln (CVFD) = 0.63 * ln (L) + 0.707, Kombolcha fibers was ln (CVFD) = 0.33 * ln (L) + 1.638, and Mekaneselam fibers was ln (CVFD) = 0.33 * ln (L) + 1.78, respectively. Variations in the cross-sectional area within the length of the fiber cause variations in stress, which may result in different overall fiber strengths. Figure 8 depicts the parameter as a measure of within-fiber diameter variation in the tested length of the measured strength. The CVFD increased with the tested length of the fiber, and the exponent was 0.47 (R-square = 90%) [35].

The shape parameter of 15 mm, 25 mm, 30 mm, 40 mm, and 50 mm tested lengths were performed based on the Weibull plot, using Equation (4). In the current result, the R^2^ coefficient range of Injibara bamboo is 94–98%, whereas Kombolcha bamboo is 89–96%, and Mekaneselam bamboo is 92–95%. This R^2^ coefficient percentage shows that they have a good linear fit and normal distribution of the strength data. The shape parameters (β) of Injibara, Kombolcha, and Mekaneselam bamboo fibers are 6.02–7.83, 5.87–10.21, and 5.86–9.63, respectively. The shape parameters show the failure strength variability, which affects the fiber’s characteristic strength. The linear regression of the fiber strength based on the tested lengths shows a minimum strength distribution variation. Kombolcha bamboo has the highest shape parameter, which is that it has a low strength distribution variation. The sensitivity parameter, shape parameter, and characteristic strength are presented in Table 2. The characteristic strength of bamboo fiber decreases as the tested length increases due to the increased likelihood of a flaw in a longer tested length. Strength variability is caused by the defect distribution within a fiber and fiber-to-fiber within a batch of fibers. The sensitivity parameters of Injibara, Kombolcha, and Mekaneselam bamboo fibers are 0.63, 0.33, and 0.33, respectively. The shape parameter (β) was used to derive the Weibull strength, whereas the sensitivity parameter (γ) was used to derive the modified Weibull strength. The Weibull and modified Weibull strengths of Injibara bamboo are high compared to Kombolcha and Mekaneselam bamboo. The current results of the shape parameter of Injibara bamboo have similar results to the previous research reported [16,25], higher than Refs. [9,24,36], and lower than Ref. [35], whereas Kombolcha bamboo is similar with Ref. [35] and higher than Refs. [9,16,24,25,36]. The current results of Injibara, Kombolcha, and Mekaneselam bamboo have higher sensitivity parameters than the previous research reported [16]. The current results of Injibara, Kombolcha, and Mekaneselam bamboo have high characteristic strength compared to previous research reported by Refs. [9,24,25], lower than Ref. [16], and similar to Ref. [36]. The sensitivity parameter (γ) of Injibara bamboo fibers is higher, whereas Kombolcha and Mekaneselam bamboo fibers have lower than previous research reported on Jute fibers [0.33] [9]. Moreover, they are lower than the previous research reported on wool fibers [0.18] [37].

### 3.5. The Significance of Diameter for Fiber Strength

The fiber strength results are presented in Table 3. The data showed that fiber strength decreased when the tested length increased. ANOVA described statistically significant strength differences between the groups of “thin” and “thick” fibers with tested lengths of 15 and 25 mm, 30 mm, 40 mm, and 50 mm. A 15 mm tested length of fibers had higher strength and a lower variance, because the elementary fibers are more likely to be clamped between the top and bottom ends. As shown in Table 4, the fibers were classified as “thin” or “thick” based on whether the fiber diameter was less than or greater than the median fiber diameter. Table 4 shows that Injibara bamboo fibers had a statistically significant difference with the group of tested lengths of 25 and 30 mm, whereas Mekaneselam bamboo fibers had a statistically significant difference with the group of testing lengths of 15 mm. However, Kombolcha bamboo showed no statistically significant difference within the group of tested lengths.

The technical bamboo fibers are composed of hundreds of elementary fibers. Consequently, even at short testing lengths, the typical discontinuities of the elementary fibers at certain points will be bridged by many other elementary fibers placed around [16]. The technical fiber strength versus their corresponding fiber diameter is indicated in Figure 8, where all samples of data are indicated. The strength of the “thin” and “thick” of Injibara and Kombolcha bamboo fibers are similar. The “thin” diameter of bamboo fibers have lower variance and higher strength than the “thick fibers” diameter. The median tensile strength for Injibara, Kombolcha, and Mekaneselam bamboo fibers was 559 MPa, 405 MPa, and 487 MPa, Whereas the median fibers diameters, were 302 μm, 263 μm, and 329 μm, respectively. As presented in Figure 8, Injibara bamboo has a lower standard deviation of strength above the median, whereas Kombolcha and Mekaneselam bamboo has a low standard deviation of strength below the median. Moreover, Injibara and Kombolcha bamboo fibers have a low standard deviation at the “thin” diameter of the fibers, whereas Mekaneselam has a low standard deviation at the “thick” diameter of fibers.

### 3.6. Measuring Tensile Strength

Table 4 shows the mean diameter, coefficient variation of fibers diameter (CVFD), tested lengths, and measured ultimate tensile strength. The CVFD was calculated as the average fiber diameter variation of 40 fibers with diameters measured at 5 mm intervals along the fiber length. The ultimate tensile strength of the fibers decreases when the tested length increases. However, the CVFD value of the bamboo fibers increases as the tested length increases. Previous studies found that within-fiber diameter variation affects fiber-breaking properties for fibers with comparable mean diameters. At long-tested lengths, a high amount of flaws and fiber diameter variations occur [9]. Table 4 shows the coefficient variation of fiber diameter of Injibara (*Y. alpina*), Kombolcha (*B. oldhamii*), and Mekaneselam (*Y. alpina*) was 11–23%, 13–20%, and 15–22%, respectively. The highest to the lowest coefficient variation of fiber diameter is recorded at Injibara (*Y. alpina*), Mekaneselam (*Y. alpina*), and Kombolcha (*B. Oldhamii*), respectively. The fiber strength for Injibara, Kombolcha, and Mekaneselam were 432–600 MPa, 370–508 MPa, and 448–566 MPa, whereas Young’s modulus was 28–50 GPa, 21–48 GPa, and 26–55 GPa, respectively. The highest to the lowest tensile strength is recorded in Injibara, Mekaneselam, and Kombolcha, whereas Young’s modulus is Mekaneselam, Injibara, and Kombolcha, respectively. The current results of tensile strength were higher compared to the previous research (205 MPa) reported [25].

### 3.7. Effect of Tested Length on the Predictive Strength

The effects of testing length on the shape and the sensitivity parameter can be seen in Table 5. The tested length influence on the modified Weibull model (Equation (5)) is presented in Table 5, along with those from the traditional Weibull model (Equation (1)) and the measured values. The measured strength and predictive Weibull strength have close data values on Injibara, Kombolcha, and Mekaneselam bamboo fibers because they have good values for the shape parameter (β) and sensitivity parameter (γ), which determine the predictive modified Weibull strength. The coefficient variation of the fiber diameter measured the predicted modified Weibull strength. However, the size of samples affects the coefficient variation of the fiber diameter at various tested lengths. If large sample sizes are used, the average diameters are a close to each other. Table 5 clearly shows that the modified model incorporates within-fiber diameter variation, which can predict the gauge length effect more accurately than the traditional Weibull distribution. Hence, based on the known fibres strength and tested length, we can predict another fiber strength based on its tested length. The predictive modified Weibull strength was significant at a very short, tested length, which cannot be achieved experimentally.

## 4. Conclusions

The quality of bamboo fibers and their composites is influenced by the extraction methods of the fibers. Currently, the extraction of bamboo fibers is a challenging task. The authors have developed a simple and easy extraction of bamboo fibers in the workshop.

At the tested lengths of 15–50 mm, the measured tensile strength of Injibara, Kombolcha, and Mekaneselam are 432 ± 59–600 ± 105 MPa, 370 ± 60–508 ± 66 MPa, and 448 ± 57–566 ± 65 MPa, whereas the characteristic tensile is 459–642 MPa, 408–638 MPa, and 474–597 MPa, and the modified Weibull strengths at the predictive tested length of 15 mm are 531–600 MPa, 489–508 MPa, and 543–566 MPa, respectively. The characteristic strength of Injibara, Kombolcha, and Mekaneselam bamboo fibers are 7%, 20%, and 5% higher than experimentally measured tensile strength. Injibara and Mekaneselam have a lower probability of failure strength compared to Kombolcha bamboo. Hence, Weibull modified strength is more closely related to the strength value with experimental strength data compared to conventional Weibull strength predictive test length at 15 mm. The sensitivity parameter (γ) of Injibara, Kombolcha, and Mekaneselam bamboo fibers was 0.63, 0.33, and 0.33, whereas the shape parameter (β) was 6.24–7.83, 5.87–10.21, and 5.86–9.63, respectively. The quality of the existing extraction machine is acceptable for the current research work, which has comparable sensitivity parameters to jute fibers as reported in the literature review. Moreover, the higher value of the shape parameter indicates that the variation of strength data is low. Further research work should be required based on the defect distribution in the area, volume, and diameter of the fibers.

## Figures and Tables

**Figure 1 materials-15-05016-f001:**
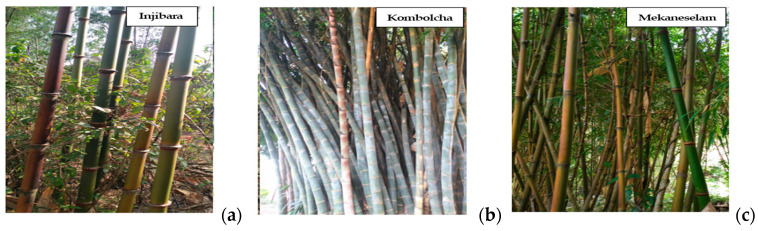
Morphological characteristics of Ethiopian bamboo species (**a**) Injibara, (**b**) Kombolcha, (**c**) and Mekaneselam.

**Figure 2 materials-15-05016-f002:**
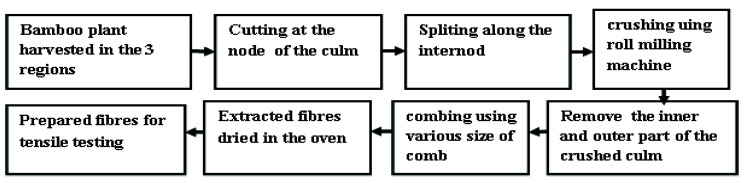
Extraction procedure of bamboo fiber.

**Figure 3 materials-15-05016-f003:**
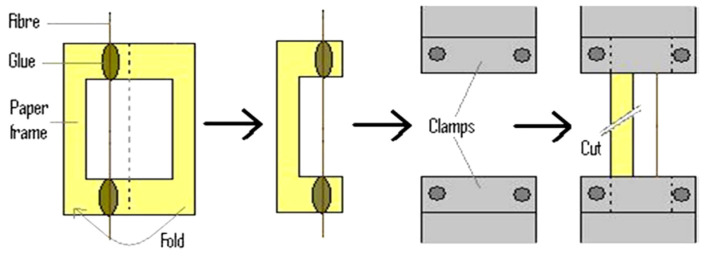
A paper frame used to keep the fiber straight on the testing machine.

**Figure 4 materials-15-05016-f004:**
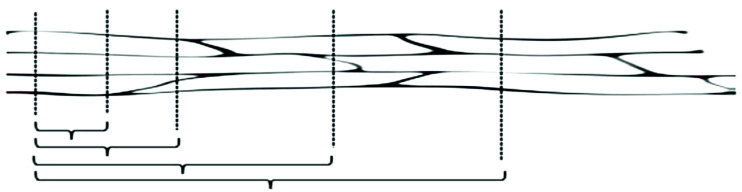
Arrangements of elementary fibers through a technical fiber that demonstrates the possibility of the fibers’ testing gauge.

**Figure 5 materials-15-05016-f005:**
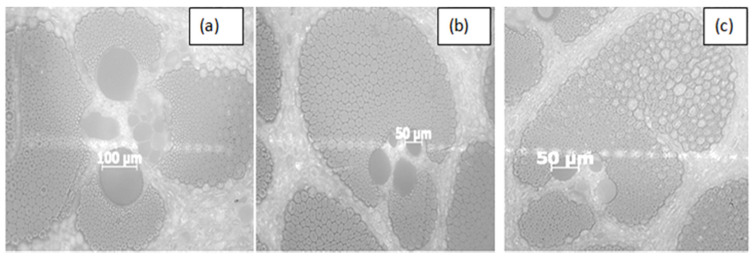
Vascular bundles of bamboo species in Ethiopia. (**a**) Injibara, (**b**) Kombolcha, (**c**) Mekaneselam bamboo.

**Figure 6 materials-15-05016-f006:**
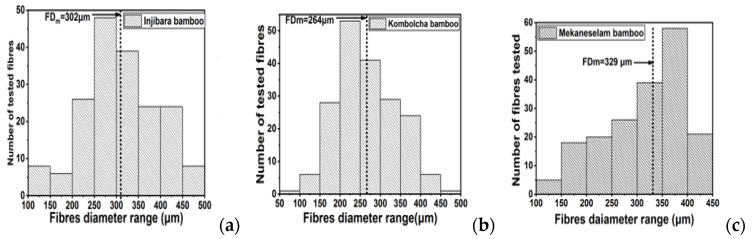
Fiber diameter (μm) distribution for all single bamboo technical samples indicating two halves of the fiber population (total number (Nt = 400)) (**a**) Injibara, (**b**) Kombolcha, and (**c**) Mekaneselam bamboo. FD_m_—fiber diameter median.

**Figure 7 materials-15-05016-f007:**
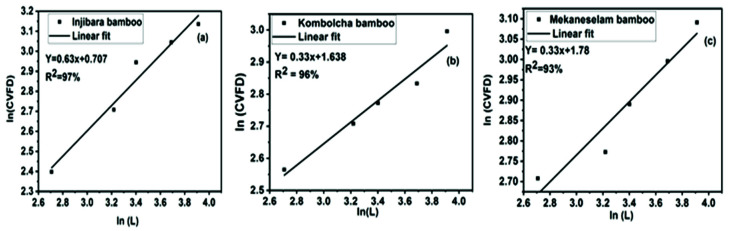
Logarithm of coefficient variation of fiber diameter versus logarithm of the tested length (**a**) Injibara, (**b**) Kombolcha, and (**c**) Mekaneselam.

**Figure 8 materials-15-05016-f008:**
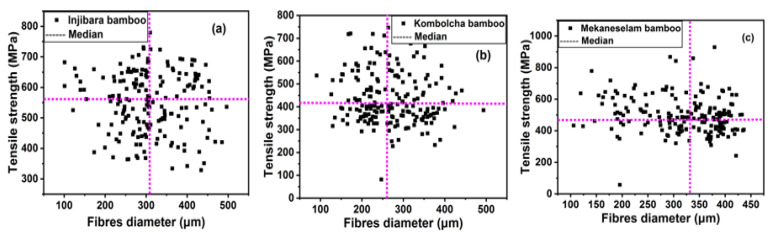
Single technical fiber strength versus fiber diameter where the median for fiber strength and fiber diameter is drawn (**a**) Injibara, (**b**) Kombolcha, and (**c**) Mekaneselam bamboo.

**Table 1 materials-15-05016-t001:** Geographic location and climatic conditions of the testing sites.

Name of Testing Sites	Administrative Location of the Testing Site				Climate, Average Value		
	Zone	Region	Lat-Long	Alt. (m)	An.RF (mm)	Max. Temp. (°C)	Min. Temp. (°C)
Injibara	Awi	Amahara	10°59′ N 36°55′ E	2540–2865	1813	24	14
Kombolcha	S/wollo	Amahara	11°5′ N 39°44′ E	1842–1915	1027	26	20
Mekaneselam	S/wollo	Amahara	10°45′ N 38°45′ E	2605–3000	1048	21	10

**Table 2 materials-15-05016-t002:** Strength of bamboo fibers using the Weibull distribution.

Bamboo Species	Gauge Length (mm)	Sensitivity Parameter (γ)	Shape Parameter (β)	Characteristic Strength (MPa)	Ref.
Injibara	15	0.63	6.24	642	
	25	0.63	7.66	617	
	30	0.63	6.40	595	study
	40	0.63	6.02	582	
	50	0.63	7.83	459	
Kombolcha	15	0.33	10.21	638	
	25	0.33	6.58	470	
	30	0.33	5.87	438	study
	40	0.33	7.32	425	
	50	0.33	7.33	408	
Mekaneselam	15	0.33	9.63	597	
	25	0.33	6.05	568	
	30	0.33	6.18	552	study
	40	0.33	5.87	488	
	50	0.33	8.36	474	
Literature	10–100	-	6.3–7.9	197–201	[25]
	5–20	-	1.19–2.18	377–436	[9]
	20–50	-	3.22–4.77	402–566	[24]
	20	-	4.02	612	[36]
	5–35	-	3.5–9.3	683–855	[35]
	1–40	0.48	7.6	982	[16]

**Table 3 materials-15-05016-t003:** Effect of various tested lengths on the tensile strength. Each tested length is divided into “thin” and “thick” fibers, which have a diameter below and above the median.

Bamboo Species	Gauge Length (mm)	Average Fiber Diameter (μm)	Average Strength All Fibers (MPa)	Average Strength “Thin” Fibers (MPa)	Average Strength “Thick” Fibers (MPa)	*p*-Value “Thin” and ”Thick” Fiber (α = 0.05)	R-Square (Linear Regression)
Injibara	15	296 ± 33	600 ± 105	602 ± 79	597 ± 128	0.876	0.0007
	25	339 ± 51	580 ± 81	512 ± 58	648 ± 23	0.000	0.7101
	30	327 ± 62	553 ± 90	560 ± 62	545 ± 75	0.000	0.6174
	40	275 ± 58	542 ± 100	563 ± 107	522 ± 91	0.207	0.0416
	50	340 ± 78	432 ± 59	431 ± 52	433 ± 68	0.916	0.0006
Kombolcha	15	253 ± 33	608 ± 66	614 ± 68	602 ± 65	0.57	0.0086
	25	248 ± 37	423 ± 83	414 ± 80	432 ± 74	0.479	0.0133
	30	256 ± 41	385 ± 73	512 ± 70	374 ± 76	0.372	0.021
	40	276 ± 47	378 ± 69	376 ± 59	380 ± 80	0.868	0.0007
	50	322 ± 64	360 ± 60	390 ± 75	386 ± 52	0.908	0.0005
Mekaneselam	15	299 ± 45	566 ± 65	590 ± 68	543 ± 55	0.022	0.1314
	25	309 ± 49	528 ± 94	531 ± 98	524 ± 94	0.817	0.0014
	30	317 ± 57	515 ± 93	527 ± 102	504 ± 86	0.442	0.0157
	40	339 ± 68	452 ± 84	454 ± 84	451 ± 86	0.925	0.0002
	50	320 ± 70	448 ± 57	438 ± 72	433 ± 54	0.903	0.0006

**Table 4 materials-15-05016-t004:** The influence of gauge length on the properties of bamboo fiber.

Bamboo Species	Gauge Length (mm)	Diameter (μm)	CVFD (%)	Fiber Strength (MPa)	Young’s Modulus (GPa)	Breaking Strain (%)	Ref.
Injibara	15	296 ± 33	11	600 ± 105	50 ± 8	1.24 ± 0.24	
	25	339 ± 51	15	580 ± 81	33 ± 5	1.45 ± 0.17	study
	30	327 ± 62	19	553 ± 90	32 ± 6	1.54 ± 0.30	
	40	275 ± 58	21	542 ± 100	30 ± 4	1.62 ± 0.25	
	50	340 ± 78	23	432 ± 59	28 ± 5	1.56 ± 0.27	
Kombolcha	15	253 ± 33	13	508 ± 66	48 ± 10	1.04 ± 0.18	
	25	248 ± 37	15	470 ± 83	38 ± 5	1.19 ± 0.24	
	30	256 ± 41	16	385 ± 73	30 ± 4	1.21 ± 0.16	study
	40	276 ± 47	17	378 ± 69	27 ± 5	1.45 ± 0.25	
	50	322 ± 64	20	400 ± 60	21 ± 3	1.57 ± 0.32	
Mekaneselam	15	299 ± 45	15	566 ± 65	55 ± 7	1.08 ± 0.25	
	25	309 ± 49	16	528 ± 94	43 ± 8	1.43 ± 0.31	
	30	317 ± 57	18	515 ± 93	42 ± 5	1.46 ± 0.27	study
	40	339 ± 68	20	452 ± 84	28 ± 5	1.59 ± 0.23	
	50	320 ± 70	22	448 ± 57	26 ± 5	1.78 ± 0.38	
Literature	10–100	24–26	9–13	201–213	-	-	[25]
	20–60	-	6–7	352–518	-	-	[24]
	1–40	132–146	-	790–943	-	-	[16]
	20–60	-	-	442–555	-	-	[36]
	5–35	366	-	639–813	33	2.0–2.9	[35]
	50	-	-	658	52	1.33	[31]

**Table 5 materials-15-05016-t005:** Summary results of the measured and the predicted strength based on the tested length (predictions are based on the results at 15 mm testing length).

Bamboo Species	Gauge Length (mm)	Measured Strength (Mpa)	Weibull Strength (Mpa)	Modified Weibull Strength (Mpa)
Injibara	15	600 ± 105	600	600
	25	580 ± 81	552	570
	30	553 ± 90	537	559
	40	542 ± 100	512	543
	50	432 ± 59	494	531
Kombolcha	15	508 ± 66	508	508
	25	470 ± 83	578	500
	30	385 ± 73	568	497
	40	378 ± 69	552	492
	50	370 ± 60	540	489
Mekaneselam	15	566 ± 65	566	566
	25	528 ± 94	537	556
	30	515 ± 93	527	553
	40	452 ± 84	511	547
	50	448 ± 57	499	543

## Data Availability

All the required data is available within the article.

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
