# Peer review of "Investigation of Bamboo Fibrous Tensile Strength Using Modified Weibull Distribution"

_materials, 2022, doi:10.3390/ma15145016_

Round 1

Reviewer 1 Report

The study presents the tensile strength of bamboo fibres representing different species collected from three different Ethiopian districts. The mechanical values are analysed using the modified Weibull analysis method. The results are interesting, they are analysed well and presented clearly in the tables. In addition, interesting comparisons are made to previous literature results for different bamboo samples.

However, the overall description of the results and the comparisons on them should have been written in a more clear way, - for example by giving the information on the bamboo species (the literature studies). Also, all the other parts of the paper do not support the results part; the introduction and the conclusions should be written in a more coherent way and they should support the results shown. Most importantly, the previous studies on Weibull analysis on various bamboo samples should be cited and presented properly already in the introduction. 

The scientific aim and scope of this study should be described more clearly to the reader; explaining what is the novelty of this study compared to the previous research. 

Besides these more general notes, the following further notes were also made, which could be checked:

*the 2nd and 4th sentences in the abstract could be clarified 

*the two first sentences of the Introduction should be checked. Also the sentences in the lines 43-45, 45-46, 47-48, 73, 77-78 must be checked and clarified.

*the Introduction should be divided into several paragraphs (so that one idea per paragraph is considered and explained in details)

*all the text should be proof read. There are some typos with capital letters, punctuation marks (e.g. lines 72, 86), and the use of verbs 

*equation 2 : the symbol sigma_f might be sigma_fi? Also the notation and explanation for n and p should be checked throughout the equations. 

*the explanations of all the equations should be clarified for the reader and the notations checked. (I would also like to recommend that only the most reasonable equations for the analysis would be shown in the paper.)

*Section 2.2: the details of the harvested bamboo species could be given with more details (the whole scientific name of the species could be given). Also references to previous studies of the properties of these species could be given; it would be very interesting.

*line 200: the error and the results should be given with as many decimals (with the same accuracy)

*please explain the acronyms in lines 206-208

*line 208: “higher” might be ‘similar as’?

*please remove the repeating expressions and sentences in line 247 (“variation within fibre diameter variation”) and 250-252

*lines 259-261: please describe further these comparisons, the species in question and consider the accuracies of the values given

*the figures (fig 8, fig 9-11, fig 12) could be printed with better resolution (the legends and the axis titles should be more readable)

*line 271: please give all the values with the same accuracy

*lines 305-306: please remove the repetition and the text could also be divided into the paragraphs to help the reader

*the conclusions could be written in a more concise way and all the numerical values given should be announced with their accuracies

*please check the reference 8

Author Response

Title: Investigation of Bamboo Fibrous Tensile Strength Using Modified Weibull Distribution

MATERIALS-ID-1765375

The authors are highly grateful to the reviewers for their constructive comments, which have helped to enhance the quality of the manuscript. Author's sincere effort has been put to revise the manuscript according to the reviewer comments. The details of revisions made in response to the comments are summarized in the following Table. The revisions in the manuscript are shown in YELLOW colored texts.

REVIEWER-COMMENT-1

AUTHOR RESPONSE

However, the overall description of the results and the comparisons on them should have been written in a more clearway, - for example by giving the information on the bamboo species (the literature studies). Also, all the other parts of the paper do not support the results part; the introduction and the conclusions should be written in a more coherent way and they should support the results shown.

Most importantly, the previous studies on Weibull analysis on various bamboo samples should be cited and presented properly already in the introduction.

Reply:

The authors appreciate the reviewer for your valuable comments to improve the quality of the manuscript in all aspects. To address the issues we thoroughly check, revised, edit, and add some content on the introduction, results, and conclusion

Explaining what is the novelty of this study compared to the previous research.

Reply: Thank you for your questions,

Ethiopia has the largest coverage of bamboo plant in Africa. However, the properties of bamboo fibres in Ethiopia has not been studied so far. The methods of fibres extraction is novel and developed by the authors. The extraction of bamboo fibres are challenged and difficult to separate the fibres from the bamboo culm. Extraction methods determine the tensile strength of the extracted fibres.

The defect distribution on the length and diameters of the fibres are affected by their morphology and methods of fibre extraction. The current research studies are considered defects distributed along the length using coefficient variance of fibres diameter to determine the strength of bamboo fibres in Ethiopia based on the developed extraction machine using modified Weibul distribution.

The aim of the current research work presents strength data of bamboo fibres in Ethiopia for an extensive population of fibres which are extracted by authors in the workshop and measured the tensile strength of the fibres based on various gauge length using UTS machine, then predict strength of fibres using modified Weibull distribution.

The 2nd and 4th sentences in the abstract could be clarified

I would like to thank for your requesting clarification,

So far, researchers have not been studied on the strength of Ethiopian bamboo fibres, which are utilized for composite applications.

The current research studies measured the strength of bamboo fibres based on various testing lengths and calculated the predictive tensile strength using a modified Weibull distribution. Moreover, the quality of the extraction machine is evaluated based on shape and sensitivity parameters.

The two first sentences of the Introduction should be checked. Also the sentences in the lines 43-45, 45-46, 47-48, 73, 77-78 must be checked and clarified.

Thank you for your suggestion, As per your comments,  I corrected and edited in sequentially.

The Introduction should be divided into several paragraphs (sothat one idea per paragraph is considered and explained indetails)

The authors appreciate your suggestion, as per your comments the introduction arranged and divided into paragraph.

All the text should be proof read. There are some typos with capital letters, punctuation marks (e.g. lines 72, 86), and the use of verbs

The authors appreciate the reviewer for your valuable comments on typos with capital lettters, punctuation marks and use of verbs to improve the quality of the manuscript in all aspects. To address the issues we thoroughly checked and edit the manuscript.

equation 2 : the symbol sigma_f might be sigma_fi? Also the notation and explanation for n and p should be checked throughout the equations.

Thank you for your suggestion, we checked and edited in a good way.

The explanations of all the equations should be clarified for the reader and the notations checked. (I would also like to recommend that only the most reasonable equations for the analysis would be shown in the paper.)

We agree with the valuable recommendation by reviewer to improve the clarity of the equation, we checked and clarified as per your comments, and some unnecessary equations are removed.

Section 2.2: the details of the harvested bamboo species could be given with more details (the whole scientific name of the species could be given). Also references to previous studies of the properties of these species could be given; it would be very interesting.

We agree with this valuable recommendation, but researchers have not studied on the properties of bamboo species in Ethiopia, so far.

Now, we have studied the whole characterization on the properties of bamboo fibres and their polymer composites for the career development of PhD

Line 200: the error and the results should be given with as many decimals (with the same accuracy)

Thank you for your comments, the number of decimal is corrected, but they have the same accuracy value.

Please explain the acronyms in lines 206-208

Thank you for your comments, as per your comments, we explain the acronyms in the manuscript.

Line 208: “higher” might be ‘similar as’?

I would like to appreciate your recommendation, We agree with this valuable recommendation.

please remove the repeating expressions and sentences in line247 (“variation within fibre diameter variation”) and 250-252

Thank you for your in-depth comments and recommendation to improve the quality, we corrected and edited

lines 259-261: please describe further these comparisons, the species in question and consider the accuracies of the values given

Thank you for your recommendation.

As you know, the accuracy is influenced by the fiber preparation, type of adhesive to attach the fibers from the paper frame, methods of extraction, and testing on the machine. For this reason, the standard deviation might be somewhat large. The standard deviation of the current strength is between 15–20%. It is normal because of the tedious procedure and the difficulty of testing single fibres. 

*the figures (fig 8, fig 9-11, fig 12) could be printed with better resolution (the legends and the axis titles should be more readable)

Thank you for your suggestion, we have performed in a good resolution.

line 271: please give all the values with the same accuracy

Thank you for your comments, the authors agree with your comments, we have corrected.

lines 305-306: please remove the repetition and the text could also be divided into the paragraphs to help the reader

Thank you for your comments, the authors agree with your comments, we have corrected.

The conclusions could be written in a more concise way and all the numerical values given should be announced with their accuracies

Thank you for your comments, the authors agree with your comments, we have arranged and written in a good way. Some numerical values could not be explained using required accuracies like characteristic strength, sensitivity parameter and shape parameters, but measured strength can be announced using their accuracy value.

*please check the reference 8

Thank you for your suggestion, I checked and replaced by latest article

Reviewer 2 Report

 Investigation of Bamboo Fibrous Tensile Strength Using Modified Weibull Distribution is an interesting publication about the cognitive nature of the properties of bamboo fibers from Ethiopia,

The Authors rightly present the structural features of bamboo fibers and studies on its microscopic structure which translates into the strength of bamboo composites.

The study was based on Weibull statistics. Evaluation based on fiber diameter was based on the assumption of the weakest link.

The tensile strength of bamboo fibers was very poorly discussed in the literature section:

Xian XJ, Xian DG (1990) The relationship of microstructure and mechanical properties of bamboo. J Bamboo Res 9(3):10–23

Yu Y, Jiang ZH, Fei BH, Wang G, Wang HK (2011) An improved microtensile technique for mechanical characterization of short plant fibers: a case study on bamboo fibers. J Mater Sci 46:739–746

Tian GL, Jiang ZH, Yu Y, Wang HK, An X (2012) Toughness mechanism of bamboo by in situ tension. J Beijing For Univ 34(5):144–147

and others

 Line 147 does not provide accurate data on test material collection statistics (diameter range and length of samples obtained.

Line 157 et seq. - no indication of what period elapsed between harvesting and the drying process and within what moisture content range of the material (did the drying process affect the mechanical changes of the fibers?)

Results are presented in an orderly and precise manner.

Lines 311 and following refer to literature values but no reference to Table 3 is given by the Authors.

The paper deserves attention but requires corrections and completing the literature review as well as referring to the studies of other Authors in the conclusions

The literature was not organized according to the requirements of the journal. Additionally there is repetition of numbers. Please refer to https://www.mdpi.com/journal/materials/instructions.

For the sake of order in the text, literature items should also be ordered.

Author Response

Title: Investigation of Bamboo Fibrous Tensile Strength Using Modified Weibull Distribution

MATERIALS-ID-1765375

The authors are highly grateful to the reviewers for their constructive comments, which have helped to enhance the quality of the manuscript. Author's sincere effort has been put to revise the manuscript according to the reviewer comments. The details of revisions made in response to the comments are summarized in the following Table. The revisions in the manuscript are shown in YELLOW colored texts.

REVIEWER-COMMENT-2

AUTHOR RESPONSE

The tensile strength of bamboo fibers was very poorly discussed in the literature section:

Xian XJ, Xian DG (1990) The relationship of microstructure and mechanical properties of bamboo. J Bamboo Res 9(3):10–23

Yu Y, Jiang ZH, Fei BH, Wang G, Wang HK (2011) An improved microtensile technique for mechanical characterization of short plant fibers: a case study on bamboo fibers. J Mater Sci 46:739–746

Tian GL, Jiang ZH, Yu Y, Wang HK, An X (2012) Toughness mechanism of bamboo by in situ tension. J Beijing For Univ34(5):144–147

Reply: Thank you for your suggestion. The authors agree with your comments, so we add some references on the introduction parts on the strength of bamboo fibres.

Line 147 does not provide accurate data on test material collection statistics (diameter range and length of samples obtained.

Reply: Thank you for your suggestion. The aim of the current research work is to measure and predict the strength based on gauge length. The diameter of the fibres is not the parameter for this study and is calculated based on the area of the fibres. The test material statistics show the candidate fibres diameter during testing. The tested fibres are selected randomly from the material collection statistics.

Line 157 et seq. - no indication of what period elapsed between harvesting and the drying process and within what moisture content range of the material (did the drying process affect the mechanical changes of the fibers?)

Reply: Thank you for your question. The moisture content is measured at the time of harvesting of the bamboo plant, but the extraction of fibres is taken at the time of harvesting without loss of its moisture content, then drying the extracted fibres at 60 ºC for 72 hrs and putting them in a climate controlled room. When we want to test the samples of the fibres are inserted at 60 ºC for 24 hrs during the measurement of the strength. The drying process might affect the mechanical properties.

Results are presented in an orderly and precise manner.

Reply: The authors appreciate the reviewer for your valuable comments on the results parts. As per your comment, the results are  arranged in orderly and  precisely.

Lines 311 and following refer to literature values but no reference to Table 3 is given by the Authors.

Reply: Thank you for your comments, but Table 3 contained the previous studies reference like 10, 17, 21, 23, 24, and 39.

The paper deserves attention but requires corrections and completing the literature review as well as referring to the studies of other Authors in the conclusions

Reply: the authors are grateful for the valuable suggestion to improve the quality of the manuscript, as per your comments, we revised and add some previous studies on the  literature reviews to improve the quality of the manuscript.

The literature was not organized according to the requirements of the journal. Additionally there is repetition of numbers. Please refer to https://www.mdpi.com/journal/materials/instructions.

Reply: We agree with the valuable comment by reviewer.

We have done revision and organized the manuscript in a good way.

For the sake of order in the text, literature items should also be ordered.

Reply: We sincerely appreciate the reviewer for the valuable comment, we ordered and organized the literature review as per your comments..

Reviewer 3 Report

Dear Authors 

I have read the manuscript about the tensile strength of bamboo fiber using the approach method of modified Weibull distribution. Many similar studies have been published before on this topic, thus it must be highlighted the difference with other related published papers. here are my comments and suggestions for improving the quality of the manuscript 

1. Most of the references must be upgraded, use only at 5 last years except some reference that they can not be replaced 

2. The introduction must be rewritten by highlighting the importance and novelty of the study compared with the updated references 

3. In the method, how to prepare the fiber is not clear, I suggest replacing figure 3 with more detail and closer insight figures. It will be better if you prepare the diagram or scheme of how you prepared the samples 

4. In the discussion you present microstructure figures, but in the method it is not mentioned 

5. Please think to reduce the number of figures, because the figure is too many in the manuscript and not clear presented 

6. The replication for each test should be mentioned 

7. Make more deep discussion for each part, not needed to discuss all in detail it will make confuse for the reader to find out the important thing that needs to point out 

8. Add some updated literature to compare your result with other papers not just presented the data in figures or in sentences, but variability in data of 3 kinds of bamboo is should be discussed, what is the reason behind the phenomena

9. Please shorten the conclusion just in the important finding for the study 

10. Adjust the abstract after revised the manuscript 

Good luck, hopefully my comments can help to improve your paper 

Author Response

Title: Investigation of Bamboo Fibrous Tensile Strength Using Modified Weibull Distribution

MATERIALS-ID-1765375

The authors are highly grateful to the reviewers for their constructive comments, which have helped to enhance the quality of the manuscript. Author's sincere effort has been put to revise the manuscript according to the reviewer comments. The details of revisions made in response to the comments are summarized in the following Table. The revisions in the manuscript are shown in YELLOW colored texts.

REVIEWER-COMMENT-3

AUTHOR RESPONSE

1# Most of the references must be upgraded, use only at 5 last year's except some reference that they can not be replaced

Reply: we agree with the valuable comment by the reviewer,  Authors revised and  add the latest reference

2# The introduction must be rewritten by highlighting the importance and novelty of the study compared with the updated references

Reply: Thank you for your suggestion, we revised and re write the introduction as per your comments.

.

3# In the method, how to prepare the fiber is not clear, I suggest replacing figure 3 with more detail and closer insight figures. It will be better if you prepare the diagram or scheme of how you prepared the samples

Reply: Thank you for your suggestion, as per your comments, we modify and change the diagram.

4# In the discussion you present microstructure figures, but in the method it is not mentioned

Reply: Thank you for your comments,

we add the procedure on the methods how to prepare the samples.

5# Please think to reduce the number of figures, because the figure is too many in the manuscript and not clear presented

Reply: Thank you for your suggestions, All figures are basic and  important to describe the reality of the research, as per your comments, we removed figure number 9,10,11 figures.

6# The replication for each test should be mentioned

Reply: The authors are grateful for the valuable suggestion, as per your comments the replication of the test are incorporate and  explained. The difficulty of testing single fibres using an UTS machine 40 samples are prepared for each test gauge length. About 200 experiments are ongoing total.

7# Make more deep discussion for each part, not needed to discuss all in detail it will make confuse for the reader to find out the important thing that needs to point out

Reply: We agree with the valuable comment by reviewer to improve the clarity for the reader. As per your comments, we revised, checked and give deep discussion in each apart to improve the quality of the manuscript.

8# Add some updated literature to compare your result with other papers not just presented the data in figures or in sentences, but variability in data of 3 kinds of bamboo is should be discussed, what is the reason behind the phenomena

Reply: We sincerely appreciate the reviewer for the valuable comment, we added some update literature to compare my result with other literature.

The current research studies characterized and measured the tensile strength of the single fibre of the 3 bamboo species in Ethiopia using an UTS machine. The three bamboo species have been harvested in different regions. Hence, they have different properties.

9#  Please shorten the conclusion just in the important finding for the study

Reply: Thank you for your suggestion.

The conclusion are shortened and revised based on your comments

10#  Adjust the abstract after revised the manuscript

Reply: I would like to appreciate your fruitful comments:

The abstract is revised and adjusted based on your comments.

Round 2

Reviewer 1 Report

The manuscript has clearly been improved by the corrections and additions.

However, the text could still be proof read (e.g. by using any automatic proof reading program) before publishing and the figure numbering should be checked (figure 1 might be missing and figures 6 & 7 should be numbered vice versa).

Author Response

Title: Investigation of Bamboo Fibrous Tensile Strength Using Modified Weibull Distribution.

MATERIALS-ID-1765375

The authors are highly grateful to the reviewers for their constructive comments, which have helped to enhance the quality of the manuscript. Author's sincere effort has been put to revise the manuscript according to the reviewer comments. The details of revisions made in response to the comments are summarized in the following Table. The revisions in the manuscript are shown in YELLOW colored texts.

REVIEWER-COMMENT-1

AUTHOR RESPONSE

However, the text could still be proof read (e.g. by using any automatic proof-reading program) before publishing and the figure numbering should be checked (figure 1 might be missing and figures 6 & 7 should be numbered vice versa).

Reply:

Thank you for your valuable comments. Sorry, we made a mistake when we revised and re ordered the whole manuscript to improve the quality of the manuscript.

 Now, I checked twice and more, then corrected the order of the figures as per your comments.

We made a proof reading of the whole manuscript and edited it in a good way.

Reviewer 3 Report

Thank you for the author's response but I suggest shortening the conclusion in fewer words but significant. This revision mode is still too long words 

Author Response

Title: Investigation of Bamboo Fibrous Tensile Strength Using Modified Weibull Distribution.

MATERIALS-ID-1765375

The authors are highly grateful to the reviewers for their constructive comments, which have helped to enhance the quality of the manuscript. Author's sincere effort has been put to revise the manuscript according to the reviewer comments. The details of revisions made in response to the comments are summarized in the following Table. The revisions in the manuscript are shown in YELLOW colored texts.

REVIEWER-COMMENT-3

AUTHOR RESPONSE

I suggest shortening the conclusion in fewer words but significant. This revision mode is still too long words

Reply: We sincerely appreciate the reviewer for the valuable comment.

As per your comments, we prepared a short and clear way of concluding
